# How internet use has transformed educational expectations in Chinese families

Chong Chen[1], Yiyan Zhang[2], Mengyuan Lu [3]*

1 School of Marxism, Zhejiang University of Technology, Hangzhou, China, 2 Wenzhou Technician Institute, Wenzhou, China, 3 School of Economics, Zhejiang University of Technology, Pingfeng Campus, Hangzhou, China

* lmy_19970@163.com

## Abstract

In this study, we investigate the impact of Internet use on educational expectations in Chinese families, utilizing social cognition theory and survey data from 2018. Our findings reveal that Internet use elevates parents' educational aspirations for their children. Furthermore, we observed that the influence of the Internet on these expectations varies and is more pronounced for older children, male parents, and families in rural areas. Besides, only using the Internet for non-entertainment purposes can promote parents' educational expectations of their children. Additionally, parents boost their children's educational expectations through enhanced social interactions facilitated by the Internet. These insights comprehensively evaluate the Internet's effect on individual perceptions and suggest strategies for promoting educational equity in the digital age.

## 1. Introduction

Educational expectations refer to parents' aspirations for their children's future education and schooling, a crucial element of family education. These expectations drive parents' involvement in the educational process, and their specific nature significantly shapes children's attitudes and ultimately their educational outcomes [1,2]. Higher family educational expectations correlate with better grades and higher educational attainment, leading to greater social status and income [3]. In the 1980s, Bourdieu [4] introduced the theory of "cultural reproduction" to explain the disparities in educational expectations among families. Based on this theory, parents with high cultural capital use cultural reproduction as a means of achieving social reproduction, thereby preserving their elite status and identity as cultural symbols.

Furthermore, Bourdieu's theory laid the foundation for subsequent research [5–8] that analyzed the varying expectations parents have for their children's education based on innate factors such as education level, occupation, and income. These studies explore the reasons why family background influences educational expectations, particularly through the provision of knowledge and material resources for

**Data availability statement:** All relevant data are within the manuscript and its Supporting Information files.

**Funding:** Zhejiang Provincial Social Science Foundation of China (Grant number 21NDJC319YBM).

**Competing interests:** The authors have declared that no competing interests exist.

further education. However, they fall short in explaining why some families with lower socio-economic status still maintain high educational aspirations for their children.

In the context of Asia, particularly China, Confucianism offers a partial explanation. In Confucian culture, education is closely linked to social status, and the ancient imperial examination system, which served as the primary means of selecting officials, reinforced the notion that education could transform family and social status. Although the imperial examination system no longer exists, the belief that disadvantaged families can achieve social mobility through education persists.

This analysis suggests that parental cognition may play a more critical role than socio-economic background in shaping educational expectations. This perspective helps explain why many students from disadvantaged backgrounds achieve high academic success, such as enrolling in major universities. While family economic and cultural strengths may remain static in the short term, parental perceptions of education's importance evolve, particularly in the Internet age where information technology is highly developed. The Internet provides an open and diverse space, liberating users from the offline world's physical and hierarchical boundaries [9]. Through the "mimetic society" mirrored by the Internet, users reshape their perceptions, emotions, and attitudes toward society [10–13]. However, the specific role of the Internet in this context remains unexplored in literature.

There are several potential correlations between Internet use and parental educational expectations. First, Internet use stimulates social learning mechanisms; parents observe and imitate role models on social media, influencing their attitudes and behaviors toward their children's education. Second, Internet use enhances social cognitive mechanisms by reducing information acquisition costs, prompting parents to update their perceptions and ultimately motivating their educational aspirations. Besides, Internet use fosters social interaction mechanisms, changing how people socialize and making it easier to connect and share feelings. Also, frequent online social interactions can heighten the psychology of comparison and the fear of falling behind, leading parents to raise their educational expectations for their children.

Furthermore, we explored the impact of the Internet on educational expectations in China. Traditionally, Chinese culture, influenced by Confucian values, places a strong emphasis on children's education. Thus, Chinese parents greatly value their children's academic success, viewing it as an honor for the entire family and even the clan [14]. Comparative studies have shown that Chinese children often outperform American children academically because Chinese parents have higher educational expectations and a stronger belief in the positive effects of individual effort [15]. Additionally, the uneven Internet penetration across Chinese regions offers a unique context for discussion. By June 2022, China had 1.051 billion Internet users, making up 74.4%(https://www.chinajusticeobserver.com/a/china-releases-50th-statistical-report-on-domestic-internet-development) of the total population, with urban penetration at 82.9% and rural penetration at 58.8%.

This study contributes significantly to research on Internet use and educational expectations. While existing literature often examines how family background factors

influence parents' educational expectations [5,6,8,16], and how Internet use affects human cognition [10–13], it rarely addresses the impact of Internet use on educational expectations specifically. We, therefore, fill this gap by investigating whether and how Internet use leads parents to raise their educational expectations for their children. The rest of this study is organized as follows: Section Two reviews the literature and develops our hypotheses; Section Three details the research design and method; Section Four presents the robustness check; and Section Five provides the summary, discussions, findings, recommendations, and the limitations of the study.

## 2. Literature review and hypothesis development

### 2.1 The digital divide and educational expectations

Recent advancements in digital technologies—centered around mobile Internet, cloud computing, and big data—have profoundly transformed economic and social development, as well as individual production and daily life. However, in the process of popularizing digital technology, significant disparities have emerged among social groups regarding access to information, the ability to acquire it, and the benefits derived from it. These differences have evolved into what is known as the "digital divide" [17].

The digital divide has a dual impact on the educational expectations of both parents and children. On one hand, unequal access to digital technology directly affects the distribution of educational resources. Viaene and Zilcha [18] argue that this inequality influences students' ability to engage with online educational resources. As digitalization progresses, an increasing number of high-quality educational resources—such as online courses, educational software, and e-books— are disseminated via the Internet. However, the digital divide prevents these resources from being equally available to all students. Urban students from affluent families are more likely to access and leverage these resources, while students from rural or economically disadvantaged backgrounds often lack the necessary technological support. This disparity in access to digital resources contributes to an educational disadvantage, which ultimately lowers the educational expectations of disadvantaged student groups.

On the other hand, the digital divide contributes to a broader family education divide' [19]. Families with higher economic capital can capitalize on their digital advantages to quickly acquire the latest and high-quality educational information, allowing them to set ambitious educational expectations for their children and support these expectations with relevant information and modern educational approaches. Conversely, families disadvantaged by the digital divide struggle to access this information and must rely on outdated or limited resources and traditional educational concepts. The resulting information asymmetry exacerbates inequalities, influencing the level and structure of these families' educational expectations for their children [20].

The impact of the digital divide on educational expectations may also be influenced by cultural backgrounds. The most typical one is gender culture. The socio-cultural norms that determine women's behavior and interests largely influence their ability to utilize information technology [21]. In cultures where gender inequality is more prominent (such as Africa, India, etc.), women may have reduced access to digital technology due to social norms [22–24]. Lack of self-worth, self-confidence, appropriate education, heavy household responsibilities, and other factors contribute to women's digital use disorders [25]. When women take on the primary responsibility for educating their children at home, the impact of the digital divide on educational inequality will be further compounded.

While the existing literature provides indirect support for the notion that the Internet impacts educational expectations through the digital divide, it lacks direct evidence with broad, policy-relevant implications.

### 2.2 Impact of Internet use on educational expectations

Based on social cognitive theory, human cognition and the environment interact continuously [26]. As users engage with the Internet, their cognition is constantly influenced by the virtual external environment they encounter. Consequently, we explain how Internet use enhances parents' educational expectations through social learning and social cognitive mechanisms.

On the one hand, it is to observe and imitate the "educational role models" in the network. The social learning mechanism highlights the dynamic relationship between the environment and individual behavior, where individuals learn appropriate behaviors by observing others. On the Internet, parents selectively observe, learn, and imitate behaviors of credible and appealing role models. For example, during online activities, users may share educational articles, and news, or showcase their educational approaches on social media to demonstrate intelligence and knowledge or to enhance their image [27–29]. These users unintentionally become role models for others. Consequently, parents observing these role models may adopt similar attitudes and behaviors. The collective emphasis of Chinese families on education is closely related to the "alternative reinforcement" in social cognitive theory. When parents observe that "role models" on the internet have gained social recognition for valuing education (such as their children being admitted to prestigious schools or their family status being elevated), they will see these results as potential rewards for their own behavior and be more actively involved in their children's education.

On the other hand, there is collectivism and the rationality of educational tools. Social cognitive mechanisms emphasize the individual's ability to interpret external information and align their self-cognition and behavior with the external environment [26]. When an individual's existing beliefs cannot accommodate new socio-cognitive information, they use self-regulation to update and reorganize their cognition to absorb and adapt to the new information. The cultural sensitivity of social cognitive theory further supports its explanatory power for Chinese family education. The collectivist tradition in Chinese society has strengthened parents' motivation to achieve "family honor" through education: firstly, the normalization of social comparison. The conspicuous behavior towards their children on social networks, such as academic performance, continues to put Chinese parents under social pressure. This comparison will trigger a vicarious experience - when parents discover the achievements of other children, they will see them as potential indicators of their own children's abilities, thereby driving up educational expectations. The second is the cognitive tool of education. The Internet in China is full of cases and views that high education brings high income. This further reinforces parents' belief that education is a tool for class transition. When parents have clear expectations for the results of their investment in education (such as future economic returns for their children), they are more willing to strengthen their educational expectations. Based on this analysis, we propose the following hypothesis:

H1: Internet use promotes parents' educational expectations for their children.

## 2.2 Mediating effects of Internet use on educational expectations for enhanced social interaction

Social interaction occurs when individual preferences, expectations, and constraints are directly influenced by the characteristics and choices of others, creating mutual influence and dependence among individuals [30]. Vygotskii and Cole [31] highlighted the crucial role social interaction plays in the development of an individual's social cognition. The Internet facilitates enhanced social interactions, allowing parents to engage with diverse perspectives and experiences. Moreover, this engagement can amplify their educational expectations by exposing them to new ideas and success stories, reinforcing the belief that education can significantly improve their children's future. In addition, the Internet influences educational expectations not only through extensive access to information but also through enhanced social interactions facilitated by social networking. Taken together, the Internet has revolutionized social interactions, removing traditional spatial and temporal constraints and strengthening previously loose social ties [32].

We argue that these social interactions indirectly impact users' educational expectations. In Asian cultures like China, where children's education is highly valued, parents often discuss their children's education during online interactions. When parents see that better educational performance leads to higher levels of satisfaction in Maslow's hierarchy of needs, such as self-esteem and self-actualization [33], they maintain high educational standards for their children. Conversely, if parents perceive their children's educational performance as lacking, the scarcity of high-quality educational resources and a competitive mentality drive them to fear "falling behind," raising their educational expectations. This

competitive comparison increases parents' anxiety about education, thus raising their expectations for their children's education. Based on this analysis, we propose the following hypothesis:

H2: The Internet increase s parents' expectations for children's education through social interaction.

## 3. Data and method

### 3.1 Data sources

We use data from the China Family Panel Studies (CFPS), produced by the China Center for Social Science Surveys at Peking University. This database, updated biennially, reflects China's economic and social development and changes in family structure. It includes variables related to individuals' Internet use, such as the amount of time spent online during leisure, and whether they access the Internet via computer or mobile device. Additionally, the data encompass individual and family characteristics relevant to our research.

Starting in the second half of 2014, China began the widespread implementation of its 4G high-speed communication network, coupled with the increasing accessibility of affordable mobile smart devices. This mobile Internet wave marked the beginning of the comprehensive influence of information technology, particularly the Internet, on various social groups. We selected 2018 survey data for our analysis from the available options. Compared to 2016, Internet development in 2018 was more advanced, with greater integration into people's daily lives. Additionally, to avoid the potential influence of the COVID-19 pandemic, we excluded data from 2020.

### 3.2 Variable settings

**3.2.1 Dependent variable.** *Edu expectation:* With China's rapid economic development and growing social competition, the demand for education has significantly increased. Thus, we use *Edu expectation* as a proxy for educational expectations in the country. A bachelor's degree or higher is often seen as a symbol of high professionalism and comprehensive ability, enhancing employment competitiveness. In the CFPS, respondents answered the question: "What level of education would you like 'children's names' to achieve? Minimum of which level of schooling?" The options ranged from elementary school to a doctorate, including high school, technical school, vocational high school, college, bachelor's degree, and master's degree. For this study, if parents expected their children to complete a bachelor's, master's, or doctoral degree, the variable was assigned a value of 1; otherwise, it was assigned a value of 0.

**3.2.2 Independent variable.** *Internet use:* In the CFPS questionnaire, a question was designed for respondents asking 'How much leisure time do you spend online per week? (hours/week)'. We take the respondent's answer to this question as a proxy variable of Internet use intensity. The average Internet use intensity is 8.585, which means that on average, parents in the sample spend more than one hour online every day. In addition, the standard deviation of this indicator is significantly greater than the average, indicating that there is a large difference in the length of Internet use between parents.

**3.2.3 Mediator variable.** *Social interaction:* There is a strong positive correlation between social gift expenditures and the level of social interaction. In traditional Chinese society, the practice of "reciprocal gift-giving" underscores the importance of social relationships. By exchanging gifts, individuals express appreciation, gratitude, and care, thereby maintaining and strengthening social bonds. As social interaction increases, individuals may incur higher expenses on gifts and favors to sustain these relationships. Since family values primarily influence offline social interactions, we measure the level of social interaction by the amount spent on social gifts, as reported in the questionnaire. In the CFPS questionnaire, a question was designed for the respondents asking "How much money did your household pay for gift giving in the last 12 months?". We measure the level of social interaction by logarithmizing the amount of personal favors and gifts paid in the questionnaire. In the sample, the households surveyed had the highest expenditure of 80000 yuan, while the households with the lowest expenditure did not.

**3.2.4 Control variable.** Based on the empirical methods utilized by previous studies, this study employed control variables for personal characteristics, family characteristics, and children's characteristics [34–36]. Personal characteristics variables encompass factors like gender and age, while household characteristics include location, income level, residency status, and household size. The control variables were chosen for the following reasons:

Child's gender. In Asian cultures, parents may be more willing to invest more resources in education for boys, who are considered to be the main breadwinners of the family. Girls may have relatively less to invest in their education, especially if resources are limited.

Child's age. The older a child is, the more stable his or her academic performance will be, and the more stable his or her parents' educational expectations will be.

Parental gender. With the spread of the concept of gender equality, the division of labour between parents tends to be more flexible. However, on the whole, mothers are more involved in their children's education and have higher expectations of their children's education.

Parental age. There is a correlation between the age of parents and their educational aspirations for their children, with younger parents usually favouring more open and diverse educational philosophies, while older parents may be more focused on traditional and stable educational paths.

Household register. Household registration is highly correlated with educational expectations. Parents with urban household registration have higher educational expectations for their children than those with agricultural registration [37].

Income. Economic advantages give parents more confidence and ability to support higher levels of education for their children, thus raising expectations for their children's education [38]. We measure the economic advantage of households by controlling for the level of household income.

Residence. Differences in the educational aspirations of rural and urban families are both a reflection of social stratification and a result of the unequal distribution of educational resources. In China, rural parents generally wish to achieve class mobility through education.

Table 1 shows the definitions and descriptive statistics of variables.

## 3.3 Model setup

Given that educational expectations are dummy variables, we used a logit model for estimation, which is able to report the dominance ratio of the variables, i.e., the ratio of the probability of an event 'occurring' to the probability of it 'not

**Table 1. Descriptive statistical analysis of variables.**

| Categorization | Variable name | Variable Definition | N | Mean | Std. Dev. |
|---|---|---|---|---|---|
| Dependent variable | *Edu expectation* | Expected years of education | 7,110 | 0.8280 | 0.3770 |
| Independent variable | *Internet use* | Internet usage time | 7,110 | 8.5850 | 11.5470 |
| Personal characteristics | *Child's gender* | Male=1, Female=0 | 7,110 | 0.5290 | 0.4990 |
| | *Child's age* | Age value | 7,110 | 7.3620 | 4.3780 |
| | *Parental gender* | Male=1, Female=0 | 7,110 | 0.3370 | 0.4730 |
| | *Parental age* | Age value | 7,110 | 41.1370 | 12.5920 |
| Family characteristics | *Household register* | Agricultural household = 1, non-agricultural household = 0 | 7,110 | 0.8080 | 0.3940 |
| | *Income* | Where does your personal income belong in your local area? 1~5 rising | 7,110 | 2.7860 | 1.1310 |
| | *Residence* | Rural = 1, Urban = 0 | 7,110 | 0.5580 | 0.4970 |
| | *Family size* | Number of family members | 7,110 | 5.4570 | 2.1630 |
| Mediator Variable | *Social interaction* | Log (Expenditures on gifts of favors +1) | 7,005 | 7.3127 | 2.3182 |

Note: Robust standard errors in parentheses; *** $p<0.01$, ** $p<0.05$, * $p<0.1$.

occurring,' and can make the probabilistic prediction problem This makes the probabilistic prediction problem more intuitive and reasonable.

Given that the dependent variable, educational expectations, is a dummy variable, we employ a Logit model to test the hypotheses. The baseline model is expressed as follow

$$Edu\ expectation_i = \beta_0 + \beta_1 Internet\ use_i + \beta_2 X_i + Area_i + \varepsilon, i = 1, 2, \ldots, n \tag{1}$$

Where, subscript $i$ represents the interviewed parent, *Edu expectation* is the dependent variable, *Internet use* is the primary explanatory variable, $X$ denotes individual and family characteristics as control variables, *Area* represents a regional dummy variable accounting for differences in educational environments, and ε is the error term.

## 4 Results

### 4.1 Baseline results

Table 2 presents the regression results of Internet use on educational expectations. Column (1) displays the estimations without control variables, while Column (2) incorporates individual child and parent characteristics variables, and Column

**Table 2. Benchmark regression results.**

| VARIABLES | (1) | (2) | (3) | (4) |
|---|---|---|---|---|
| | Logit | Logit | Logit | Odd Ratio |
| *Internet use* | 0.0141*** | 0.0143*** | 0.0111*** | 1.0111*** |
| | (0.0032) | (0.0037) | (0.0036) | (0.0036) |
| *Child's gender* | | 0.0006 | 0.0030 | 1.0030 |
| | | (0.0634) | (0.0638) | (0.0640) |
| *Child's age* | | −0.0562*** | −0.0556*** | 0.9459*** |
| | | (0.0075) | (0.0076) | (0.0071) |
| *Parental gender* | | −0.2423*** | −0.2659*** | 0.7665*** |
| | | (0.0673) | (0.0678) | (0.0519) |
| *Parental age* | | 0.0083*** | 0.0078** | 1.0078** |
| | | (0.0031) | (0.0031) | (0.0031) |
| *Household register* | | | −0.8159*** | 0.4422*** |
| | | | (0.1012) | (0.0448) |
| *Income* | | | −0.0204 | 0.9798 |
| | | | (0.0288) | (0.0282) |
| *Residence* | | | 0.0738 | 1.0766 |
| | | | (0.0654) | (0.0704) |
| *Family size* | | | −0.0048 | 0.9952 |
| | | | (0.0146) | (0.0146) |
| Area Control | √ | √ | √ | √ |
| Constant | 1.5468*** | 1.7167*** | 1.4430*** | 11.7496*** |
| | (0.0601) | (0.1522) | (0.1170) | (2.4625) |
| Observations | 7,110 | 7,110 | 7,110 | 7,110 |

Note: Robust standard errors in parentheses;

***$p<0.01$,

**$p<0.05$,

*$p<0.1$.

(3) further includes family-related control variables. In columns (1)–(3), the regression coefficients of Internet use are significantly positive at the level of 1%, which indicates that the longer parents use the Internet, the greater the probability that they expect their children to go to four-year colleges. From then on, it is assumed that H1 has been validated. As the coefficients of the Logit model lack practical significance and solely indicate significance and direction, odds ratios and marginal effects were computed. Columns (4) sequentially present the odds ratios and average effects based on the estimation results in Column (3). For each additional hour of weekly Internet use, parents' expectations of their children attending undergraduate college increase by 1.11%.

Regarding odds ratios, as children grow older, parents' expectations of them attending a four-year college decrease by 5.41% after controlling for variables. Parents typically hold higher expectations for younger children in the earlier stages of education. Male parents are 23.35% less likely than female parents to expect their children to attend a four-year college, a reflection of China's gendered family division of labor, where fathers often bear financial responsibilities while mothers prioritize family matters. The odds ratio for parental age is 1.0078, indicating that with each additional year of parental age, the odds of expecting their children to attend undergraduate school increase by 0.78%. An older parent often implies greater family resources to support their child's education, including better opportunities, stable finances, and increased cultural capital. Rural parents with registered residence are 44.22% less likely than urban parents to expect their children to attend a four-year college, attributable to economic and social disparities between urban and rural families and the concentration of rural students in schools with inferior instruction quality.

## 4.2 Robustness check

### 4.2.1 Consideration of Internet usage patterns.
Using cell phones to access the Internet: With the proliferation of smartphones and comprehensive Internet infrastructure, accessing the Internet via cell phones has become increasingly prevalent. As of 2023, the total number of Chinese cell phone users has reached 1.079 billion, with 99.8% accessing the Internet through their phones. Table 3, Column (1), indicates that parents who access the Internet via cell phones hold higher educational expectations for their children compared to those who don't.

Use of computers to access the Internet: Computers represent a more traditional means of Internet access, requiring higher user literacy. Table 3, Column (2), reveals that using computers to access the Internet also enhances parents' educational expectations, aligning with previous findings.

### 4.2.2 Estimated higher education.
As education gains prominence and social competition intensifies in China, the once-dominant competitive edge of a bachelor's degree in employment opportunities begins to wane. Consequently, it's unsurprising that some parents now aspire for their children to pursue graduate degrees. To gauge the impact of Internet use on educational expectations, we utilized the question "Do parents expect their children to attend graduate school (master's or doctoral)?" The variable is coded as 1 if the respondent anticipates their child completing a master's or doctoral degree, and 0 otherwise. The results in Table 3, Column (3), demonstrate a positive and significant effect of Internet use on the expectation for children to attend graduate school.

Furthermore, with only 12.25% of parents in the sample expecting their children to pursue graduate studies, there's a risk of "rare event bias" in the estimation results of the traditional Logit model. To address this, we performed a re-estimation using the complementary log-log model (*cloglog*). Column (4) presents the results of the *cloglog* model based on Column (3), yielding consistent findings.

### 4.2.3 Addressing endogenous issues.

(1) IV-Probit model

Endogeneity issues may arise in the regression model due to several factors. Firstly, omitted variables, such as the family's attitude toward risk or their inclination to accept new concepts, could impact both Internet use and parental educational expectations. Secondly, some families inherently harbor high educational expectations for their children, and the

**Table 3. Robustness test results.**

| VARIABLES | (1) Via computer | (2) Via Mobile phone | (3) Master | (4) Cloglog |
|---|---|---|---|---|
| *Computer* | 0.6712*** | | | |
| | (0.1015) | | | |
| *Mobile phone* | | 0.2402*** | | |
| | | (0.0822) | | |
| *Internet use* | | | 0.0065** | 0.0062** |
| | | | (0.0033) | (0.0030) |
| *Child's gender* | 0.0031 | 0.0049 | 0.1853** | 0.1746** |
| | (0.0640) | (0.0638) | (0.0735) | (0.0685) |
| *Child's age* | −0.0546*** | −0.0570*** | −0.0356*** | −0.0330*** |
| | (0.0076) | (0.0076) | (0.0089) | (0.0082) |
| *Parental gender* | −0.3135*** | −0.2780*** | −0.0820 | −0.0733 |
| | (0.0684) | (0.0681) | (0.0814) | (0.0746) |
| *Parental age* | 0.0087*** | 0.0094*** | 0.0087*** | 0.0082*** |
| | (0.0029) | (0.0035) | (0.0032) | (0.0030) |
| *Household register* | −0.6644*** | −0.8017*** | −0.5702*** | −0.5276*** |
| | (0.1056) | (0.1024) | (0.0850) | (0.0771) |
| *Income* | −0.0236 | −0.0191 | 0.0100 | 0.0111 |
| | (0.0285) | (0.0285) | (0.0327) | (0.0307) |
| *Residence* | 0.0760 | 0.0743 | 0.0440 | 0.0449 |
| | (0.0655) | (0.0654) | (0.0748) | (0.0695) |
| *Family size* | −0.0028 | −0.0050 | −0.0101 | −0.0101 |
| | (0.0146) | (0.0146) | (0.0166) | (0.0161) |
| Area Control | √ | √ | √ | √ |
| Constant | 2.2560*** | 2.3377*** | −1.8918*** | −1.9800*** |
| | (0.2022) | (0.2397) | (0.2111) | (0.1994) |
| Observations | 7,110 | 7,110 | 7,110 | 7,110 |

Note: Robust standard errors in parentheses;

***$p<0.01$,

**$p<0.05$,

*$p<0.1$.

demand for Internet use could stem from a desire for learning, such as information retrieval or online classes, leading to a reverse causation dilemma. To tackle potential endogeneity problems, we employ the instrumental variables method. Due to the inability of the Logit model to directly apply the instrumental variable method, following the practices of most studies, the IV-Probit model is adopted for instrumental variable estimation.

We utilize a dummy variable representing household ownership of a computer in 2014 as an instrument for Internet use. The presence of a computer in the household is strongly correlated with Internet usage. If a household owned a computer in 2014, its members likely continued to use the Internet in subsequent years, potentially transitioning from computer to mobile Internet use. Importantly, household computer ownership is unrelated to parental educational expectations, meeting the exogeneity condition of the instrumental variable. Re-estimation using the IV-Probit model is conducted, with the results presented in Table 4. Column (1) displays the first-stage regression outcomes of the IV-probit estimation, demonstrating a significant positive effect of computer ownership in 2014 on Internet use. This validates the correlation of

**Table 4. Endogeneity test results.**

| VARIABLES | (1) | (2) | (3) | (4) |
|---|---|---|---|---|
| | Internet use | IV-probit | Placebo | PSM |
| *Internet use* | | 0.0869*** | | 0.0184** |
| | | (0.0182) | | (0.0079) |
| *Owned Computer* | 2.5568*** | | | |
| | (0.2644) | | | |
| *Placebo* | | | −0.0007 | |
| | | | (0.0028) | |
| Child's gender | 0.1211 | −0.0024 | 0.0030 | −0.0246 |
| | (0.2413) | (0.0423) | (0.0638) | (0.1126) |
| Child's age | −0.0241 | −0.0268*** | −0.0565*** | −0.0587*** |
| | (0.0289) | (0.0051) | (0.0076) | (0.0131) |
| Parental gender | 0.9612*** | −0.2224*** | −0.2580*** | −0.3064** |
| | (0.2601) | (0.0478) | (0.0677) | (0.1220) |
| Parental age | −0.3610*** | 0.0339*** | 0.0038 | 0.0199*** |
| | (0.0102) | (0.0070) | (0.0028) | (0.0063) |
| Household register | −1.9808*** | −0.2386*** | −0.8463*** | −0.6818*** |
| | (0.3197) | (0.0775) | (0.1006) | (0.2148) |
| Income | −0.3527*** | 0.0247 | −0.0237 | 0.0094 |
| | (0.1085) | (0.0198) | (0.0287) | (0.0510) |
| Residence | 0.0699 | 0.0464 | 0.0729 | 0.1694 |
| | (0.2461) | (0.0431) | (0.0653) | (0.1137) |
| Family size | 0.0348 | −0.0110 | −0.0044 | −0.0151 |
| | (0.0566) | (0.0099) | (0.0146) | (0.0267) |
| Area Control | √ | √ | √ | √ |
| Constant | 25.0057 | −0.7709 | 2.7719*** | 1.7986*** |
| | (0.7116) | (0.5155) | (0.1885) | (0.4236) |
| Wald test | 137.20*** | | | |
| F test | 166.75*** | | | |
| Observations | 6,478 | 6,478 | 7,110 | 2,141 |

*Note*: Robust standard errors in parentheses;

***$p<0.01$,

**$p<0.05$,

*$p<0.1$; The Wald test result of the IV-probit model is chi2(1) = 137.20 with $p$-value 0.00, accepting the original hypothesis of exogeneity of instrumental variables; the F-test result is higher than the critical value of 10, indicating no problem of weak instrumental variables.

the instrumental variable and its explanatory power for the endogenous variable. Column (2) showcases the results of the second-stage regression estimated by IV-probit, revealing a significant coefficient of Internet use (0.0869) at the 1% level.

(2) Placebo testing

Reverse numerical order assignment was employed for the placebo test. Specifically, we utilized reverse alphabetical assignment to disrupt the intensity of parents' Internet use. For instance, the ID of the first parent in our sample when listed in numerical order, 100724601, was erroneously assigned 987204431. The results, depicted in Column (3) of Table 4, indicate that the coefficient on Internet use is not significant, which demonstrates the robustness of the benchmark results.

(3) PSM model testing

Parents' decisions regarding Internet use are often non-randomized. To tackle the endogeneity problem arising from the "self-selection" of individuals' decisions to use the Internet, this study employs the propensity score matching (PSM) method to estimate the treatment effect of Internet use on educational expectations. Parents who use the Internet constitute the experimental group, while parents who do not use the Internet form the control group. A 1:1 matching was executed utilizing the nearest neighbor matching method, considering variables such as parent's gender, age, household location, income level, and place of residence. As shown in Column (4) of Table 4, the conclusion that Internet use fosters educational aspirations persists even after employing the propensity score matching method to address the "self-selection" bias in Internet use.

## 4.3 Educational expectations based on different Internet use purposes

We further examine how different online activities influence educational expectations. The CFPS database includes a survey that asks individuals about the frequency of their Internet use for various purposes, including learning, work, business, socializing, and entertainment. Respondents' frequency of use is rated on a scale: almost every day=7, 3–4 times a week=6, 1–2 times a week=5, 2–3 times a month=4, once a month=3, once every few months=2, and never=1. The results are presented in Table 5.

Our findings reveal that when parents use the Internet for learning and work, their educational expectations for their children are the highest (Coef=0.1211, p<0.01; Coef=0.1170, p<0.01), followed by business activities and social intercourse (Coef=0.1015, p<0.01; Coef=0.0825, p<0.01). As expected, frequent online entertainment by parents does not significantly enhance their expectations for their children's education (Coef=−0.0002, p>0.1). Compared to parents who primarily engage in recreational activities online, those who use the Internet for studying, working, and conducting business are more likely to gain knowledge and self-improvement, which reinforces their focus on their children's education. These results indicate that the purpose of Internet use can significantly impact parents' educational expectations.

## 4.4 Heterogeneity analysis

The empirical results in Table 2 demonstrate that parental educational expectations for their children are significantly influenced by factors such as the child's age, the parent's gender, and the family's residence, owing to the dynamics of family division of labor and social cognition. Building on the theoretical analysis in the preceding section, which posits that the impact of Internet use is stronger on socially disadvantaged parents, it is inferred that the effect of the Internet on educational expectations varies across different groups. Utilizing Equation (1), we segmented the sample into groups based on parental residence, parental gender, and children's schooling stage, with results presented in Table 6.

Columns (1–2) unveil the outcomes of tests conducted on household residence groups. In the rural group, the regression coefficient for Internet use is 0.0195 and significant; similarly, in the urban group, it is 0.0173 and significant, indicating noteworthy differences in the coefficients between the groups for Internet use. This suggests a more pronounced contribution of Internet use to educational aspirations among parents in rural households, emphasizing its significant impact on improving educational expectations among socially disadvantaged parents. Columns (3–4) present the findings based on tests for parental gender grouping. The regression coefficient for Internet use is 0.0244 and significant in the male parent group, while in the female parent group, it is 0.0163 and significant, demonstrating a significant difference in the coefficients between the groups for Internet use. This implies a greater influence of Internet use on the educational expectations of male parents compared to female parents. Columns (5–7) depict the results of subgroup tests based on different educational stages (elementary, middle, and high school) of the children. The findings suggest that Internet use can alter parents' educational expectations, even for children in higher grades. To ensure robustness, we further tested for significant differences in coefficients between groups. The results of the Seemingly Unrelated Regression (SUEST)

**Table 5. The purpose of internet use and educational expectations.**

| VARIABLES | (1) | (2) | (3) | (4) | (5) |
|---|---|---|---|---|---|
| | Logit | Logit | Logit | Logit | Logit |
| *Study* | 0.1211*** | | | | |
| | (0.0197) | | | | |
| *Work* | | 0.1170*** | | | |
| | | (0.0208) | | | |
| *Business* | | | 0.1015*** | | |
| | | | (0.0226) | | |
| *Social intercourse* | | | | 0.0825*** | |
| | | | | (0.0237) | |
| *Entertainment* | | | | | −0.0002 |
| | | | | | (0.0219) |
| *Child's gender* | −0.0521 | −0.1852* | −0.0523 | −0.0522 | −0.0554 |
| | (0.0831) | (0.0968) | (0.0830) | (0.0830) | (0.0828) |
| *Child's age* | −0.0562*** | −0.0637*** | −0.0569*** | −0.0578*** | −0.0582*** |
| | (0.0103) | (0.0118) | (0.0104) | (0.0104) | (0.0103) |
| *Parental gender* | −0.3795*** | −0.3919*** | −0.3288*** | −0.3211*** | −0.3222*** |
| | (0.0896) | (0.0992) | (0.0887) | (0.0892) | (0.0887) |
| *Parental age* | 0.0195*** | 0.0167** | 0.0207*** | 0.0180*** | 0.0141** |
| | (0.0060) | (0.0070) | (0.0060) | (0.0061) | (0.0060) |
| *Household register* | −0.8235*** | −0.8496*** | −0.8460*** | −0.9107*** | −0.9389*** |
| | (0.1240) | (0.1534) | (0.1242) | (0.1216) | (0.1216) |
| *Income* | 0.0268 | 0.0716 | 0.0359 | 0.0439 | 0.0468 |
| | (0.0377) | (0.0499) | (0.0374) | (0.0377) | (0.0377) |
| | 0.1728** | 0.2034** | 0.1610* | 0.1534* | 0.1563* |
| | (0.0847) | (0.0982) | (0.0846) | (0.0845) | (0.0844) |
| *Family size* | −0.0047 | 0.0031 | −0.0050 | −0.0070 | −0.0046 |
| | (0.0192) | (0.0226) | (0.0190) | (0.0191) | (0.0191) |
| Area Control | √ | √ | √ | √ | √ |
| Constant | 1.7514*** | 1.8795*** | 1.6816*** | 1.7130*** | 2.3507*** |
| | (0.3032) | (0.3706) | (0.3265) | (0.3433) | (0.3288) |
| Observations | 4,587 | 3,447 | 4,588 | 4,585 | 4,586 |

Note: Robust standard errors in parentheses;

***$p<0.01$,

**$p<0.05$,

*$p<0.1$; The deletion of samples that answered 'I don't know' or were missing resulted in the missing sample size in Table 5.

analysis revealed that the p-values for the differences in regression coefficients across groups were all less than 0.01, indicating significant variations in the regression coefficients between the groups. To present the heterogeneity analysis more clearly, we plot the results in Fig 1.

The preceding observations suggest that the Internet exerts a more pronounced influence on demographics less receptive and aware of information, particularly among younger age groups and individuals with lower educational attainment and gender disparities [39–41].

**Table 6. Heterogeneity analysis results.**

| VARIABLES | (1) Primary school | (2) Junior high school | (3) Senior high school | (4) Male parents | (5) Female parents | (6) Living in the city | (7) Living in the countryside |
|---|---|---|---|---|---|---|---|
| *Internet use* | **0.0042** | **0.0193*** | **0.0262**** | **0.0145**** | **0.0101**** | **0.0054** | **0.0165*** |
| | **(0.0045)** | **(0.0064)** | **(0.0116)** | **(0.0064)** | **(0.0042)** | **(0.0050)** | **(0.0050)** |
| *Child's gender* | 0.0918 | −0.0176 | −0.2295 | −0.0206 | 0.0080 | −0.0713 | 0.0697 |
| | (0.0941) | (0.1013) | (0.1753) | (0.1054) | (0.0804) | (0.0954) | (0.0861) |
| *Child's age* | −0.0762*** | −0.0269 | 0.2122 | −0.0588*** | −0.0538*** | −0.0601*** | −0.0518*** |
| | (0.0221) | (0.0297) | (0.1576) | (0.0125) | (0.0097) | (0.0114) | (0.0103) |
| *Parental gender* | −0.3188*** | −0.2164** | −0.2362 | / | / | −0.3297*** | −0.2077** |
| | (0.1015) | (0.1061) | (0.1792) | | | (0.1011) | (0.0915) |
| *Parental age* | 0.0058 | 0.0133*** | −0.0005 | 0.0102** | 0.0073* | 0.0080* | 0.0075* |
| | (0.0042) | (0.0048) | (0.0086) | (0.0047) | (0.0039) | (0.0045) | (0.0040) |
| *Household register* | −0.9535*** | −0.6971*** | −0.6120** | −0.8636*** | −0.7932*** | −0.7887*** | −0.8536*** |
| | (0.1537) | (0.1576) | (0.2644) | (0.1592) | (0.1308) | (0.1439) | (0.1422) |
| *Income* | −0.0269 | −0.0247 | 0.0110 | −0.1185** | 0.0283 | 0.0631 | −0.0879** |
| | (0.0417) | (0.0450) | (0.0743) | (0.0492) | (0.0346) | (0.0426) | (0.0380) |
| *Residence* | 0.0689 | 0.0675 | 0.1523 | 0.1226 | 0.0570 | / | / |
| | (0.0960) | (0.1029) | (0.1755) | (0.1070) | (0.0822) | | |
| *Family size* | 0.0081 | −0.0176 | −0.0144 | 0.0111 | −0.0156 | −0.0239 | 0.0078 |
| | (0.0225) | (0.0236) | (0.0396) | (0.0247) | (0.0189) | (0.0236) | (0.0195) |
| Area Control | √ | √ | √ | √ | √ | √ | √ |
| Constant | 2.7000*** | 1.8634*** | −1.1992 | 2.2931*** | 2.4029*** | 2.3702*** | 2.6201*** |
| | (0.2969) | (0.4484) | (2.3682) | (0.3428) | (0.2619) | (0.3017) | (0.2795) |
| Observations | 3,736 | 2,603 | 771 | 2,398 | 4,712 | 3,142 | 3,968 |

Note: Robust standard errors in parentheses;

***$p<0.01$,

**$p<0.05$,

*$p<0.1$.

## 4.5 Mechanism testing

China epitomizes a society built on human relationships, where communication channels have evolved from traditional letters to modern telephones. In contemporary times, social networking platforms facilitated by the Internet have emerged as pivotal tools for daily interaction, offering advantages such as freedom from temporal and spatial constraints, feature-rich interfaces, and user-friendly operation, thereby fostering seamless communication. Building on findings by McKenna and Bargh [42] suggesting that individuals perceive online communication as more uninhibited than face-to-face interaction, thereby fostering social relationships online that extend into real life, our inquiry aims to ascertain whether Internet use indeed encourages interpersonal interaction within households.

Given that the Logit regression model is a nonlinear probability model, this study utilizes the Kaplan Meier with Breslow (KHB) method for tiered data to test the mediating effects within such models. The KHB method decomposes the results into three components: the estimation results of the simplified (Reduced) model, the full (Full) model, and the difference (Diff) between the two. Moreover, the simplified model represents the total effect, while the full model represents the direct effect, and the difference between the two reflects the mediating effect.

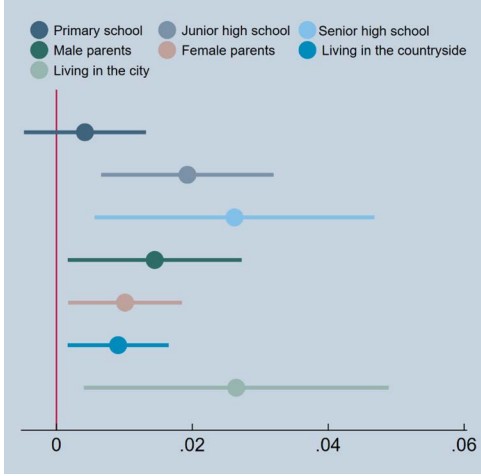

**Fig 1. Comparison of coefficients for heterogeneity analysis.**

Table 7 presents the decomposition results of the mediating effect of social interaction on the relationship between Internet use and educational expectations. The findings indicate that the estimated coefficient of the difference model is 0.0005, significant at the 1% level, suggesting that social interaction mediates the positive impact of Internet use on educational expectations. This result implies that Internet use enhances parents' social interactions, and through these interactions, parents acquire more information (shaping their cognition) or raise their educational expectations for their children, driven by comparative psychology.

## 5. Discussion

### 5.1 Summary

This study delves into two main aspects. Firstly, we explore the intersection of household Internet access and parental educational expectations, which diverges from previous studies primarily focused on the influence of Internet use on children's academic performance [43]. Contrary to conventional wisdom, we emphasize the impact of parents' perceptions shaped by Internet usage on their educational expectations for their children [16]. This shift in perspective underscores the intricate relationship between parental perceptions, children's self-expectations, and academic outcomes, as emphasized by Gonçalves, Rocha [44]. Our findings reveal new ways of linking Internet use to children's educational outcomes through parental expectations, and find that Chinese parents will be more likely to expect their children to go to college under the influence of the Internet. Of course, this influence is not always valid across cultural contexts. On the one hand, parents from Asian backgrounds will have higher academic standards and higher expectations for their children's education [45]. The cognitive and learning mechanisms brought about by the Internet may be weaker for parents from cultures with low expectations for their children's education (e.g., completion of an apprenticeship). As they are not concerned

**Table 7. Internet use for educational expectations test - Social interaction effects.**

| Model | Coefficient | Standard error | Z value | P value |
|---|---|---|---|---|
| Reduced Model | 0.0103 | 0.0035 | 2.96 | 0.003 |
| Full Model | 0.0097 | 0.0035 | 2.81 | 0.005 |
| Diff Model | 0.0005 | 0.0002 | 2.52 | 0.012 |

with their children's education, parents may be keen to share their children's life entertainment rather than educational achievements, which may also lead to the failure of the imitation and role modelling effects that we have described in our hypothesis. On the other hand, the notion of collectivism is also an important driver. Children's achievements are often seen as the success of the family, and children's efforts and progress can bring positive social evaluation and status enhancement to the family. For cultures that emphasise individualism, parents may be more concerned with their own needs, which in turn weakens the expectation of family status enhancement through their children.

Second, we extend our inquiry to provide theoretical insights into existing research paradigms. Our analysis reveals that the influence of Internet use on educational aspirations intensifies with the advancement of a child's educational stage, particularly affecting the aspirations of male parents and families residing in rural areas. These nuanced findings surpass the scope of previous studies, which predominantly examined age differences among children or variations in family economic conditions. While the above goes beyond the studies of some scholars who have focused mainly on differences in children's age [46], or family economic conditions [47], our study adopts a more comprehensive perspective encompassing child, parent, and family dynamics, thereby enriching the discourse on the heterogeneous effects of Internet use on educational expectations.

Thirdly, our analysis refines the delineation between the spheres of the Internet and educational expectations, whereas prior scholars have tended to lump the Internet together as a collective entity—encompassing various devices [48] and focused primarily on parents' educational expectations for children below the undergraduate level [49]. On one hand, we scrutinized how the duration and type of Internet usage influence educational expectations. Our research results emphasize that accessing the Internet via mobile phones or computers can affect parents' expectations of children's education. Nevertheless, it must be for non-entertainment purposes. On the other hand, we investigated whether parents aspire for their children to pursue master's or doctoral degrees. The results revealed a positive correlation between parents' extended Internet use and their desire for their children to attain higher academic qualifications. These findings contribute significantly to their respective research domains.

Fourthly, our analyses unveil that Internet use not only impacts parents' educational expectations for their children but also influences the dynamics of interactions between parents and society. This discovery aligns with existing research highlighting the nexus between social interactions and individuals' social cognition[31,50]. We particularly focus on the mechanisms of societal interaction primarily through household expenditures. Empirical evidence validates that Internet use escalates household spending on favors and overall borrowing and lending, corroborating our earlier hypothesis.

## 5.2 Findings, policy recommendations, and research limitations

We elucidate the impact of Internet use on educational expectations, drawing from social cognitive theory and data gleaned from Chinese parents. The primary findings of our study are as follows: Firstly, the more time parents dedicate to Internet usage weekly, the greater the likelihood they expect their children to attend a four-year college. Secondly, Internet use indirectly influences parental educational expectations for their children through bolstered social interaction effects. Besides, the effect of the Internet on parental educational expectations exhibits heterogeneity. We observe that this effect is more pronounced when: (1) Children are younger rather than older; (2) Male parents instead of female parents are involved; and (3) Families reside in rural areas as opposed to urban locales.

Exploring the impact of Internet use on educational expectations holds significant relevance. Educational inequity persists in countries worldwide, with families from diverse economic backgrounds holding varying educational ideals [16,51]. Consequently, enhancing educational equity remains a paramount objective in global education development efforts. Our study empirically illustrates that Internet usage can transcend inherent cognitive limitations across diverse populations, fostering shifts in educational expectations. Therefore, harnessing the Internet as a catalyst for educational change should emerge as a pivotal driver for sustainable education development in China and globally. To this end, we give the following policy recommendations: first, expand rural broadband network coverage and reduce the cost of access for households

in remote areas, in response to the more significant impact of the Internet in rural households. Second, subsidise the purchase of smart terminals (e.g., tablet PCs) for low-income people to ensure hardware accessibility. Finally, create officially certified online education communities to facilitate the sharing of experiences between urban and rural parents.

That said, this study has several limitations. First, the digital divide encompasses three levels: "access gap," "usage gap," and "knowledge gap." However, due to data constraints, this study primarily focuses on the impact of "access channels" on educational expectations, leaving the other dimensions unexplored. Second, the limited number of years does not allow for estimating the effect of the long-term impact of internet use on educational aspirations. Third, although China's unique educational climate and regionally differentiated Internet penetration make it an ideal case for analyzing the Internet's impact on educational expectations, our empirical results are specific to China. Cross-country comparative analyses can subsequently be used to test the cultural universality and institutional boundary conditions of the Internet's influence on educational expectations.

## Supporting information

**S1 File.**
(DO)

**S2 File.**
(DTA)

## Author contributions

**Conceptualization:** Chong Chen.

**Data curation:** Chong Chen.

**Formal analysis:** Chong Chen.

**Investigation:** Yiyan Zhang.

**Methodology:** Yiyan Zhang.

**Project administration:** Yiyan Zhang.

**Resources:** MengYuan Lu.

**Software:** MengYuan Lu.

**Supervision:** Yiyan Zhang.

**Validation:** MengYuan Lu.

**Visualization:** Yiyan Zhang.

**Writing – original draft:** Chong Chen.

**Writing – review & editing:** Chong Chen.

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
