## [Decision Letter · Decision Letter 0]

30 Jul 2024

PONE-D-24-23805How Internet Use Has Transformed Educational Expectations in Chinese FamiliesPLOS ONE

Dear Dr. Lu,

Thank you for submitting your manuscript to PLOS ONE. After careful consideration, we feel that it has merit but does not fully meet PLOS ONE’s publication criteria as it currently stands. Therefore, we invite you to submit a revised version of the manuscript that addresses the points raised during the review process.

Thank you for submitting your manuscript titled "How Internet Use Has Transformed Educational Expectations in Chinese Families " to PLOS ONE. After a thorough review by our expert reviewers, I have decided that the paper requires major revisions before it can be considered for publication. Below, I summarize the key points from the reviewers' comments and provide specific recommendations for addressing them.

Your study addresses an important and intriguing question regarding the educational expectations of parents in China. The reviewers have provided valuable feedback to enhance the clarity, robustness, and theoretical grounding of your manuscript. I encourage you to carefully address each of the points raised and resubmit your revised manuscript.

We look forward to receiving your revised manuscript and appreciate your contributions to the field.

We look forward to receiving your revised manuscript.

Kind regards,

Wenbin Du

Academic Editor

PLOS ONE

 [Zhejiang Provincial Social Science Foundation of China (Grant number 21NDJC319YBM)].  

**Comments to the Author**

1. Is the manuscript technically sound, and do the data support the conclusions?

Reviewer #1: Yes

Reviewer #2: Yes

2. Has the statistical analysis been performed appropriately and rigorously? 

Reviewer #1: Yes

Reviewer #2: No

3. Have the authors made all data underlying the findings in their manuscript fully available?

Reviewer #1: Yes

Reviewer #2: Yes

4. Is the manuscript presented in an intelligible fashion and written in standard English?

Reviewer #1: Yes

Reviewer #2: Yes

5. Review Comments to the Author

Reviewer #1: This paper addressed an interesting question: why parents hold high education expectations in China (even for those from low SES backgrounds)? As we know, low SES parents in other countries like U.S. usually maintain low education expectations instead. The authors argued that internet use might be a reason. I have several comments on this paper.

First, the authors should elaborate more on how and why internet use affect education expectations, particularly in Literature Review and Heterogeneity analysis. Also, the measurement is "the spare time for internet", but I think it is necessary to differentiate types of usage, like whether they use internet for Tik-tok, self-learning or business. The second-level and third-level of digital divide has argued the importance of usage and gain via internet. (The authors may also add some literature on digital divide.) If CFPS lacks such measurement, the authors should explain it and admit such limitation.

Second, hypothesis 2, the social interaction mechanism, needs revision. If the authors wanted to test the mechanism, they'd better use mediation analysis. Currently, we only see the correlation between internet use and social interaction in Table 6, yet how social interaction, particularly the measurements in this paper, affect education expectations is not discussed.

Third, I would like to propose another comment on the theoretical framework. We usually use Confucian culture to explain the high education expectations of the Chinese. Yet if the high education expectations are driven by internet use, how can we explain the high education expectations before the wide access to internet since 1990s? I think the authors should reconsider this question and revise the Introduction accordingly.

Reviewer #2: The paper provides a comprehensive overview of the research area and is well-structured. This is an interesting study and the authors have used a unique dataset and cuttingedge methodology to demonstrate it. However, the paper can be enhanced by addressing the following aspects:

1.Longitudinal Data and Cross-Sectional Analysis:

The China Family Panel Studies (CFPS) data used in this article is longitudinal, meaning it follows the same individuals over time. However, in your study, the authors pooled data from different waves (2016, 2018, and 2020) and treated it as cross-sectional data. This approach has limitations because the measuremts are not independent of each other in different waves.

To improve your analysis, consider either:

Using a specific round of data (e.g., 2016) for cross-sectional analysis.

Implementing panel data analysis to account for the three waves of data and individual-level changes.

2. Model Consistency and Terminology:

In Section 3.3, the authors describe the benchmark model as a logit model (p17), but the header in Table 2 (P19) refers to it as probit. Consistency is crucial.

Additionally, the header in Column (4) of Table 2 should be “odds ratio,” not “rate.” (P19) It should be noticed that ratios compare homogeneous indicators, while rates compare heterogeneous indicators.

3. Robustness test between logit and probit:

While logit and probit models are almost the same link functions, there is a very rough approximation between the coefficients of the probit and logit: multiply the probit coefficient by 1.8 to get logit coefficient, this can be traced back to the (pi/sqrt(3)) difference in the variances of the models. Therefore, the probit model in the first column of table 3 should not be used as the robustness test result of the logit model.(P21)

4. Heterogeneity Tests and Chow Test:

When all heterogeneity test coefficients in Table 5 are significant (p25), it doesn’t necessarily mean one group’s coefficient is greater than another’s.

To make such claims, consider using the Chow test. It assesses whether coefficients differ significantly between groups, providing more robust evidence of differential impacts.

6. PLOS authors have the option to publish the peer review history of their article (what does this mean? ). If published, this will include your full peer review and any attached files.

**Do you want your identity to be public for this peer review?** For information about this choice, including consent withdrawal, please see our Privacy Policy .

Reviewer #1: No

Reviewer #2: No

---

## [Author Response · Author response to Decision Letter 1]

26 Aug 2024

Response to Reviewers' Comments

Manuscript ID: PONE-D-24-23805

How Internet Use Has Transformed Educational Expectations in Chinese Families

Thank you for the opportunity to revise and resubmit the manuscript (PONE-D-24-23805 ) for Plos One. We sincerely appreciate the reviewers' comments. We consider all the comments an opportunity to improve the manuscript. We have now addressed all the suggested comments. The detailed response is tabulated in the following section. We have marked all the revised parts in red for your convenience.

Reviewer 1

This paper addressed an interesting question: why parents hold high education expectations in China (even for those from low SES backgrounds)? As we know, low SES parents in other countries like U.S. usually maintain low education expectations instead. The authors argued that internet use might be a reason. I have several comments on this paper.

Response:

We would like to thank you for your very positive feedback and are grateful to you for your constructive and insightful comments. We have addressed all your comments and now outline our point-by-point responses below.

Reviewer 1 - Comment 1

The authors should elaborate more on how and why internet use affect education expectations, particularly in Literature Review and Heterogeneity analysis. Also, the measurement is "the spare time for internet", but I think it is necessary to differentiate types of usage, like whether they use internet for Tik-tok, self-learning or business. The second-level and third-level of digital divide has argued the importance of usage and gain via internet. (The authors may also add some literature on digital divide.) If CFPS lacks such measurement, the authors should explain it and admit such limitation.

Response 1:

Thank you for this specific comment. We have made three modifications: First, we further analyzed the impact of different Internet use purposes (Study, Work, Business, Social intercourse, Entertainment) on educational expectations in the empirical study; Secondly, we have added literature and theories on the digital divide and educational expectations in the literature review and theoretical hypothesis sections; Thirdly, we discussed the limitations of the research based on the three levels of digital divide.Please see below for what we have revised:

In section Abstract (on page: 1):

… ' Besides, only using the Internet for non-entertainment purposes can promote parents' educational expectations of their children. '…

In section 2.1 The digital divide and educational expectations (on pages: 4-6):

2.1 The digital divide and educational expectations

Recent advancements in digital technologies—centered around mobile Internet, cloud computing, and big data—have profoundly transformed economic and social development, as well as individual production and daily life. However, in the process of popularizing digital technology, significant disparities have emerged among social groups regarding access to information, the ability to acquire it, and the benefits derived from it. These differences have evolved into what is known as the "digital divide" (17).

The digital divide has a dual impact on the educational expectations of both parents and children. On one hand, unequal access to digital technology directly affects the distribution of educational resources. Viaene and Zilcha (18) argue that this inequality influences students' ability to engage with online educational resources. As digitalization progresses, an increasing number of high-quality educational resources—such as online courses, educational software, and e-books—are disseminated via the Internet. However, the digital divide prevents these resources from being equally available to all students. Urban students from affluent families are more likely to access and leverage these resources, while students from rural or economically disadvantaged backgrounds often lack the necessary technological support. This disparity in access to digital resources contributes to an educational disadvantage, which ultimately lowers the educational expectations of disadvantaged student groups.

On the other hand, the digital divide contributes to a broader 'family education divide' (19). Families with higher economic capital can capitalize on their digital advantages to quickly acquire the latest and high-quality educational information, allowing them to set ambitious educational expectations for their children and support these expectations with relevant information and modern educational approaches. Conversely, families disadvantaged by the digital divide struggle to access this information and must rely on outdated or limited resources and traditional educational concepts. The resulting information asymmetry exacerbates inequalities, influencing the level and structure of these families' educational expectations for their children (20).

While the existing literature provides indirect support for the notion that the Internet impacts educational expectations through the digital divide, it lacks direct evidence with broad, policy-relevant implications.

In section 4.3 Educational expectations under different Internet use purposes (on pages: 17-19):

4.3 Educational expectations based on different Internet use purposes

We further examine how different online activities influence educational expectations. The CFPS database includes a survey that asks individuals about the frequency of their Internet use for various purposes, including learning, work, business, socializing, and entertainment. Respondents' frequency of use is rated on a scale: almost every day=7, 3-4 times a week=6, 1-2 times a week=5, 2-3 times a month=4, once a month=3, once every few months=2, and never=1. The results are presented in Table 5.

Our findings reveal that when parents use the Internet for learning and work, their educational expectations for their children are the highest, followed by business activities and social networking. As expected, frequent online entertainment by parents does not significantly enhance their expectations for their children's education. Compared to parents who primarily engage in recreational activities online, those who use the Internet for studying, working, and conducting business are more likely to gain knowledge and self-improvement, which reinforces their focus on their children's education. These results indicate that the purpose of Internet use can significantly impact parents' educational expectations.

Table 5 The Purpose of Internet Use and Educational Expectations

In section 5.1 Summary (on pages: 22-25):

… ' Our research results emphasize that accessing the Internet via mobile phones or computers can affect parents' expectations of children's education. Nevertheless, it must be for non-entertainment purposes. '…

… ' That said, this study has several limitations. First, the digital divide encompasses three levels: "access gap," "usage gap," and "knowledge gap." However, due to data constraints, this study primarily focuses on the impact of "access channels" on educational expectations, leaving the other dimensions unexplored.'…

Reviewer 1 - Comment 2

Hypothesis 2, the social interaction mechanism, needs revision. If the authors wanted to test the mechanism, they'd better use mediation analysis. Currently, we only see the correlation between internet use and social interaction in Table 6, yet how social interaction, particularly the measurements in this paper, affect education expectations is not discussed.

Response 2:

Thank you very much for this comment. We further strengthened the measurement of social interaction and reexamined the mechanism. Please see below for what we have revised:

In section 3.2.3 Mediator Variable (on page: 10):

Social interaction: There is a strong positive correlation between social gift expenditures and the level of social interaction. In traditional Chinese society, the practice of "reciprocal gift-giving" underscores the importance of social relationships. By exchanging gifts, individuals express appreciation, gratitude, and care, thereby maintaining and strengthening social bonds. As social interaction increases, individuals may incur higher expenses on gifts and favors to sustain these relationships. Since family values primarily influence offline social interactions, we measure the level of social interaction by the amount spent on social gifts, as reported in the questionnaire.

In section 4.5 Mechanism testing (on pages: 21-22):

Given that the Logit regression model is a nonlinear probability model, this study utilizes the Kaplan Meier with Breslow (KHB) method for tiered data to test the mediating effects within such models. The KHB method decomposes the results into three components: the estimation results of the simplified (Reduced) model, the full (Full) model, and the difference (Diff) between the two. Moreover, the simplified model represents the total effect, while the full model represents the direct effect, and the difference between the two reflects the mediating effect.

Table 7 presents the decomposition results of the mediating effect of social interaction on the relationship between Internet use and educational expectations. The findings indicate that the estimated coefficient of the difference model is 0.0005, significant at the 1% level, suggesting that social interaction mediates the positive impact of Internet use on educational expectations. This result implies that Internet use enhances parents' social interactions, and through these interactions, parents acquire more information (shaping their cognition) or raise their educational expectations for their children, driven by comparative psychology.

Table 7 Internet Use for Educational Expectations Test - Social Interaction Effects

Reviewer 1 - Comment 3

I would like to propose another comment on the theoretical framework. We usually use Confucian culture to explain the high education expectations of the Chinese. Yet if the high education expectations are driven by internet use, how can we explain the high education expectations before the wide access to internet since 1990s? I think the authors should reconsider this question and revise the Introduction accordingly.

Response 3:

Thank you very much for this comment. We have revised the introduction, especially emphasizing the reasons for high education expectations before the advent of the Internet. Please see below for what we have revised:

In section Introduction (on pages: 1-2):

… ' In the 1980s, Bourdieu (4) introduced the theory of "cultural reproduction" to explain the disparities in educational expectations among families. Based on this theory, parents with high cultural capital use cultural reproduction as a means of achieving social reproduction, thereby preserving their elite status and identity as cultural symbols.

Furthermore, Bourdieu's theory laid the foundation for subsequent research (5-8) that analyzed the varying expectations parents have for their children's education based on innate factors such as education level, occupation, and income. These studies explore the reasons why family background influences educational expectations, particularly through the provision of knowledge and material resources for further education. However, they fall short in explaining why some families with lower socio-economic status still maintain high educational aspirations for their children.

In the context of Asia, particularly China, Confucianism offers a partial explanation. In Confucian culture, education is closely linked to social status, and the ancient imperial examination system, which served as the primary means of selecting officials, reinforced the notion that education could transform family and social status. Although the imperial examination system no longer exists, the belief that disadvantaged families can achieve social mobility through education persists.

This analysis suggests that parental cognition may play a more critical role than socio-economic background in shaping educational expectations. '…

Reviewer 2

The paper provides a comprehensive overview of the research area and is well-structured. This is an interesting study and the authors have used a unique dataset and cuttingedge methodology to demonstrate it. However, the paper can be enhanced by addressing the following aspects

Response:

We would like to thank you for your very positive feedback and are grateful to you for your constructive and insightful comments. We have addressed all your comments and outlined our point-by-point responses below.

Reviewer 2 - Comment 1

Longitudinal Data and Cross-Sectional Analysis: The China Family Panel Studies (CFPS) data used in this article is longitudinal, meaning it follows the same individuals over time. However, in your study, the authors pooled data from different waves (2016, 2018, and 2020) and treated it as cross-sectional data. This approach has limitations because the measuremts are not independent of each other in different waves.

To improve your analysis, consider either:Using a specific round of data (e.g., 2016) for cross-sectional analysis.Implementing panel data analysis to account for the three waves of data and individual-level changes.

Response 1:

Thank you for this specific comment. We used data from 2018 for cross-sectional analysis and made changes to the research results of the entire article, while the research conclusions remained largely unchanged. See below for what we have added:

In section 3.1 Data sources (on page: 9):

… ' We selected 2018 survey data for our analysis from the available options. Compared to 2016, Internet development in 2018 was more advanced, with greater integration into people's daily lives. Additionally, to avoid the potential influence of the COVID-19 pandemic, we excluded data from 2020.

In section 3.2.3 Control variable (on page: 11):

Table 1 Descriptive statistical analysis of variables

Reviewer 2 - Comment 2

Model Consistency and Terminology:In Section 3.3, the authors describe the benchmark model as a logit model (p17), but the header in Table 2 (P19) refers to it as probit. Consistency is crucial.Additionally, the header in Column (4) of Table 2 should be “odds ratio,” not “rate.” (P19) It should be noticed that ratios compare homogeneous indicators, while rates compare heterogeneous indicators.

Response2:

Thank you for this specific comment. We have corrected the wording errors by replacing probit with logit and changing "odd rate" to "odd ratio". See below for what we have added:

In section 4.1 Baseline results (on page: 13):

Table 2 Benchmark regression results

Reviewer 2 - Comment 3

Robustness test between logit and probit:While logit and probit models are almost the same link functions, there is a very rough approximation between the coefficients of the probit and logit: multiply the probit coefficient by 1.8 to get logit coefficient, this can be traced back to the (pi/sqrt(3)) difference in the variances of the models. Therefore, the probit model in the first column of table 3 should not be used as the robustness test result of the logit model.(P21)

Response3:

Thank you for this specific comment. We removed the probit model from the robustness test.

Reviewer 2 - Comment 4

Heterogeneity Tests and Chow Test: When all heterogeneity test coefficients in Table 5 are significant (p25), it doesn’t necessarily mean one group’s coefficient is greater than another’s. To make such claims, consider using the Chow test. It assesses whether coefficients differ significantly between groups, providing more robust evidence of differential impacts.

Response4:

Thank you for this specific comment. We used Seemingly Unrelated Regression (SUEST) to confirm that there were differences in coefficients between the groups. See below for what we have added:

In section 4.4 Heterogeneity analysis (on pages: 19-20):

… ' To ensure robustness, we further tested for significant differences in coefficients between groups. The results of the Seemingly Unrelated Regression (SUEST) analysis revealed that the p-values for the differences in regression coefficients across groups were all less than 0.01, indicating significant variations in the regression

---

## [Editor Report · Decision Letter 1]

1 Oct 2024

PONE-D-24-23805R1How Internet Use Has Transformed Educational Expectations in Chinese FamiliesPLOS ONE

Dear Dr. Lu,

Thank you for submitting your manuscript to PLOS ONE. After careful consideration, we feel that it has merit but does not fully meet PLOS ONE’s publication criteria as it currently stands. Therefore, we invite you to submit a revised version of the manuscript that addresses the points raised during the review process.

Thank you for submitting your manuscript, "How Internet Use Has Transformed Educational Expectations in Chinese Families," to PLOS ONE. After careful review by two experts in the field, I have evaluated their comments and feedback, and I regret to inform you that your manuscript has been deemed in need of major revisions before it can be considered for publication. I understand that these revisions may require substantial effort, but I believe they are essential to enhancing the quality and impact of your work. I encourage you to carefully consider each of the reviewers' comments and suggestions, and to work diligently to address them in a thorough and thoughtful manner. Please note that this major revision request does not necessarily preclude eventual publication in PLOS ONE. However, it is crucial that you respond to the reviewers' concerns in a satisfactory manner. I recommend that you submit a detailed response to the reviewers' comments along with your revised manuscript, highlighting the changes you have made and explaining how they address the reviewers' concerns. I look forward to receiving your revised manuscript and evaluating it for publication. If you have any questions or need further guidance, please do not hesitate to contact me. 

We look forward to receiving your revised manuscript.

Kind regards,

Wenbin Du

Academic Editor

PLOS ONE

---

## [Author Response · Author response to Decision Letter 2]

14 Oct 2024

Response to Reviewers' Comments

Manuscript ID: PONE-D-24-23805

How Internet Use Has Transformed Educational Expectations in Chinese Families

Thank you for the opportunity to revise and resubmit the manuscript (PONE-D-24-23805 ) for Plos One. We sincerely appreciate the reviewers' comments. We consider all the comments an opportunity to improve the manuscript. We have now addressed all the suggested comments. The detailed response is tabulated in the following section. We have marked all the revised parts in red for your convenience.

Reviewer 1

This paper addressed an interesting question: why parents hold high education expectations in China (even for those from low SES backgrounds)? As we know, low SES parents in other countries like U.S. usually maintain low education expectations instead. The authors argued that internet use might be a reason. I have several comments on this paper.

Response:

We would like to thank you for your very positive feedback and are grateful to you for your constructive and insightful comments. We have addressed all your comments and now outline our point-by-point responses below.

Reviewer 1 - Comment 1

The authors should elaborate more on how and why internet use affect education expectations, particularly in Literature Review and Heterogeneity analysis. Also, the measurement is "the spare time for internet", but I think it is necessary to differentiate types of usage, like whether they use internet for Tik-tok, self-learning or business. The second-level and third-level of digital divide has argued the importance of usage and gain via internet. (The authors may also add some literature on digital divide.) If CFPS lacks such measurement, the authors should explain it and admit such limitation.

Response 1:

Thank you for this specific comment. We have made three modifications: First, we further analyzed the impact of different Internet use purposes (Study, Work, Business, Social intercourse, Entertainment) on educational expectations in the empirical study; Secondly, we have added literature and theories on the digital divide and educational expectations in the literature review and theoretical hypothesis sections; Thirdly, we discussed the limitations of the research based on the three levels of digital divide.Please see below for what we have revised:

In section Abstract (on page: 1):

… ' Besides, only using the Internet for non-entertainment purposes can promote parents' educational expectations of their children. '…

In section 2.1 The digital divide and educational expectations (on pages: 4-6):

2.1 The digital divide and educational expectations

Recent advancements in digital technologies—centered around mobile Internet, cloud computing, and big data—have profoundly transformed economic and social development, as well as individual production and daily life. However, in the process of popularizing digital technology, significant disparities have emerged among social groups regarding access to information, the ability to acquire it, and the benefits derived from it. These differences have evolved into what is known as the "digital divide" (17).

The digital divide has a dual impact on the educational expectations of both parents and children. On one hand, unequal access to digital technology directly affects the distribution of educational resources. Viaene and Zilcha (18) argue that this inequality influences students' ability to engage with online educational resources. As digitalization progresses, an increasing number of high-quality educational resources—such as online courses, educational software, and e-books—are disseminated via the Internet. However, the digital divide prevents these resources from being equally available to all students. Urban students from affluent families are more likely to access and leverage these resources, while students from rural or economically disadvantaged backgrounds often lack the necessary technological support. This disparity in access to digital resources contributes to an educational disadvantage, which ultimately lowers the educational expectations of disadvantaged student groups.

On the other hand, the digital divide contributes to a broader 'family education divide' (19). Families with higher economic capital can capitalize on their digital advantages to quickly acquire the latest and high-quality educational information, allowing them to set ambitious educational expectations for their children and support these expectations with relevant information and modern educational approaches. Conversely, families disadvantaged by the digital divide struggle to access this information and must rely on outdated or limited resources and traditional educational concepts. The resulting information asymmetry exacerbates inequalities, influencing the level and structure of these families' educational expectations for their children (20).

While the existing literature provides indirect support for the notion that the Internet impacts educational expectations through the digital divide, it lacks direct evidence with broad, policy-relevant implications.

In section 4.3 Educational expectations under different Internet use purposes (on pages: 17-19):

4.3 Educational expectations based on different Internet use purposes

We further examine how different online activities influence educational expectations. The CFPS database includes a survey that asks individuals about the frequency of their Internet use for various purposes, including learning, work, business, socializing, and entertainment. Respondents' frequency of use is rated on a scale: almost every day=7, 3-4 times a week=6, 1-2 times a week=5, 2-3 times a month=4, once a month=3, once every few months=2, and never=1. The results are presented in Table 5.

Our findings reveal that when parents use the Internet for learning and work, their educational expectations for their children are the highest, followed by business activities and social networking. As expected, frequent online entertainment by parents does not significantly enhance their expectations for their children's education. Compared to parents who primarily engage in recreational activities online, those who use the Internet for studying, working, and conducting business are more likely to gain knowledge and self-improvement, which reinforces their focus on their children's education. These results indicate that the purpose of Internet use can significantly impact parents' educational expectations.

Table 5 The Purpose of Internet Use and Educational Expectations

VARIABLES (1) (2) (3) (4) (5)

Logit Logit Logit Logit Logit

Study 0.1211***

(0.0197)

Work 0.1170***

(0.0208)

Business 0.1015***

(0.0226)

Social intercourse 0.0825***

(0.0237)

Entertainment -0.0002

(0.0219)

Child’s gender -0.0521 -0.1852* -0.0523 -0.0522 -0.0554

(0.0831) (0.0968) (0.0830) (0.0830) (0.0828)

Child’s age -0.0562*** -0.0637*** -0.0569*** -0.0578*** -0.0582***

(0.0103) (0.0118) (0.0104) (0.0104) (0.0103)

Parental gender -0.3795*** -0.3919*** -0.3288*** -0.3211*** -0.3222***

(0.0896) (0.0992) (0.0887) (0.0892) (0.0887)

Parental age 0.0195*** 0.0167** 0.0207*** 0.0180*** 0.0141**

(0.0060) (0.0070) (0.0060) (0.0061) (0.0060)

Household register -0.8235*** -0.8496*** -0.8460*** -0.9107*** -0.9389***

(0.1240) (0.1534) (0.1242) (0.1216) (0.1216)

Income 0.0268 0.0716 0.0359 0.0439 0.0468

(0.0377) (0.0499) (0.0374) (0.0377) (0.0377)

0.1728** 0.2034** 0.1610* 0.1534* 0.1563*

(0.0847) (0.0982) (0.0846) (0.0845) (0.0844)

Family size -0.0047 0.0031 -0.0050 -0.0070 -0.0046

(0.0192) (0.0226) (0.0190) (0.0191) (0.0191)

Area Control √ √ √ √ √

Constant 1.7514*** 1.8795*** 1.6816*** 1.7130*** 2.3507***

(0.3032) (0.3706) (0.3265) (0.3433) (0.3288)

Observations 4,587 3,447 4,588 4,585 4,586

Note: Robust standard errors in parentheses; *** p<0.01, ** p<0.05, * p<0.1� The deletion of samples that answered 'I don't know' or were missing resulted in the missing sample size in Table 5.

In section 5.1 Summary (on pages: 22-25):

… ' Our research results emphasize that accessing the Internet via mobile phones or computers can affect parents' expectations of children's education. Nevertheless, it must be for non-entertainment purposes. '…

… ' That said, this study has several limitations. First, the digital divide encompasses three levels: "access gap," "usage gap," and "knowledge gap." However, due to data constraints, this study primarily focuses on the impact of "access channels" on educational expectations, leaving the other dimensions unexplored.'…

Reviewer 1 - Comment 2

Hypothesis 2, the social interaction mechanism, needs revision. If the authors wanted to test the mechanism, they'd better use mediation analysis. Currently, we only see the correlation between internet use and social interaction in Table 6, yet how social interaction, particularly the measurements in this paper, affect education expectations is not discussed.

Response 2:

Thank you very much for this comment. We further strengthened the measurement of social interaction and reexamined the mechanism. Please see below for what we have revised:

In section 3.2.3 Mediator Variable (on page: 10):

Social interaction: There is a strong positive correlation between social gift expenditures and the level of social interaction. In traditional Chinese society, the practice of "reciprocal gift-giving" underscores the importance of social relationships. By exchanging gifts, individuals express appreciation, gratitude, and care, thereby maintaining and strengthening social bonds. As social interaction increases, individuals may incur higher expenses on gifts and favors to sustain these relationships. Since family values primarily influence offline social interactions, we measure the level of social interaction by the amount spent on social gifts, as reported in the questionnaire.

In section 4.5 Mechanism testing (on pages: 21-22):

Given that the Logit regression model is a nonlinear probability model, this study utilizes the Kaplan Meier with Breslow (KHB) method for tiered data to test the mediating effects within such models. The KHB method decomposes the results into three components: the estimation results of the simplified (Reduced) model, the full (Full) model, and the difference (Diff) between the two. Moreover, the simplified model represents the total effect, while the full model represents the direct effect, and the difference between the two reflects the mediating effect.

Table 7 presents the decomposition results of the mediating effect of social interaction on the relationship between Internet use and educational expectations. The findings indicate that the estimated coefficient of the difference model is 0.0005, significant at the 1% level, suggesting that social interaction mediates the positive impact of Internet use on educational expectations. This result implies that Internet use enhances parents' social interactions, and through these interactions, parents acquire more information (shaping their cognition) or raise their educational expectations for their children, driven by comparative psychology.

Table 7 Internet Use for Educational Expectations Test - Social Interaction Effects

Model Coefficient Standard error Z value P value

Reduced Model 0.0103 0.0035 2.96 0.003

Full Model 0.0097 0.0035 2.81 0.005

Diff Model 0.0005 0.0002 2.52 0.012

Reviewer 1 - Comment 3

I would like to propose another comment on the theoretical framework. We usually use Confucian culture to explain the high education expectations of the Chinese. Yet if the high education expectations are driven by internet use, how can we explain the high education expectations before the wide access to internet since 1990s? I think the authors should reconsider this question and revise the Introduction accordingly.

Response 3:

Thank you very much for this comment. We have revised the introduction, especially emphasizing the reasons for high education expectations before the advent of the Internet. Please see below for what we have revised:

In section Introduction (on pages: 1-2):

… ' In the 1980s, Bourdieu (4) introduced the theory of "cultural reproduction" to explain the disparities in educational expectations among families. Based on this theory, parents with high cultural capital use cultural reproduction as a means of achieving social reproduction, thereby preserving their elite status and identity as cultural symbols.

Furthermore, Bourdieu's theory laid the foundation for subsequent research (5-8) that analyzed the varying expectations parents have for their children's education based on innate factors such as education level, occupation, and income. These studies explore the reasons why family background influences educational expectations, particularly through the provision of knowledge and material resources for further education. However, they fall short in explaining why some families with lower socio-economic status still maintain high educational aspirations for their children.

In the context of Asia, particularly China, Confucianism offers a partial explanation. In Confucian culture, education is closely linked to social status, and the ancient imperial examination system, which served as the primary means of selecting officials, reinforced the notion that education could transform family and social status. Although the imperial examination system no longer exists, the belief that disadvantaged families can achieve social mobility through education persists.

This analysis suggests that parental cognition may play a more critical role than socio-economic background in shaping educational expectations. '…

Reviewer 2

The paper provides a comprehensive overview of the research area and is well-structured. This is an interesting study and the authors have used a unique dataset and cuttingedge methodology to demonstrate it. However, the paper can be enhanced by addressing the following aspects

Response:

We would like to thank you for your very positive feedback and are grateful to you for your constructive and insightful comments. We have addressed all your comments and outlined our point-by-point responses below.

Reviewer 2 - Comment 1

Longitudinal Data and Cross-Sectional Analysis: The China Family Panel Studies (CFPS) data used in this article is longitudinal, meaning it follows the same individuals over time. However, in your study, the authors pooled data from different waves (2016, 2018, and 2020) and treated it as cross-sectional data. This approach has limitations because the measuremts are not independent of each other in different waves.

To improve your analysis, consider either:Using a specific round of data (e.g., 2016) for cross-sectional analysis.Implementing panel data analysis to account for the three waves of data and individual-level changes.

Response 1:

Thank you for this specific comment. We used data from 2018 for cross-sectional analysis and made changes to the research results of the entire article, while the research conclusions remained largely unchanged. See below for what we have added:

In section 3.1 Data sources (on page: 9):

… ' We selected 2018 survey data for our analysis from the available options. Compared to 2016, Internet development in 2018 was more advanced, with greater integration into people's daily lives. Additionally, to avoid the potential influence of the COVID-19 pandemic, we excluded data from 2020.

In section 3.2.3 Control variable (on page: 11):

Table 1 Descriptive statistical analysis of variables

Categorization Variable name Variable Definition N Mean Std. Dev.

Dependent variable Edu expectation Expected years of education 7,110 0.8280 0.3770

Independent variable Internet use Internet usage time 7,110 8.5850 11.5470

Personal characteristics Child’s gender Male=1, Female=0 7,110 0.5290 0.4990

Child’s age Age value 7,110 7.3620 4.3780

Parental gender Male=1, Female=0 7,110 0.3370 0.4730

Parental age Age value 7,110 41.1370 12.5920

Family characteristics Household register Agricultural household = 1, non-agricultural household = 0 7,110 0.8080 0.3940

Income Where does your personal income belong in your local area? 1~5 rising 7,110 2.7860 1.13

---

## [Decision Letter · Decision Letter 2]

22 Jan 2025

PONE-D-24-23805R2How Internet Use Has Transformed Educational Expectations in Chinese FamiliesPLOS ONE

Dear Dr. Lu,

Thank you for submitting your manuscript to PLOS ONE. After careful consideration, we feel that it has merit but does not fully meet PLOS ONE’s publication criteria as it currently stands. Therefore, we invite you to submit a revised version of the manuscript that addresses the points raised during the review process.

Dear authors, Please see the comments and revise your paper accordingly. Good LuckBest,Academic Editor ==============================

We look forward to receiving your revised manuscript.

Kind regards,

Ehsan Namaziandost

Academic Editor

PLOS ONE

Reviewers' comments:

Reviewer's Responses to Questions

**Comments to the Author**

1. If the authors have adequately addressed your comments raised in a previous round of review and you feel that this manuscript is now acceptable for publication, you may indicate that here to bypass the “Comments to the Author” section, enter your conflict of interest statement in the “Confidential to Editor” section, and submit your "Accept" recommendation.

Reviewer #3: (No Response)

2. Is the manuscript technically sound, and do the data support the conclusions?

Reviewer #3: Yes

3. Has the statistical analysis been performed appropriately and rigorously? 

Reviewer #3: Yes

4. Have the authors made all data underlying the findings in their manuscript fully available?

Reviewer #3: (No Response)

5. Is the manuscript presented in an intelligible fashion and written in standard English?

Reviewer #3: (No Response)

6. Review Comments to the Author

Reviewer #3: Dear Editor,

I would like to express my gratitude for the opportunity to review the manuscript titled "How Internet Use Has Transformed Educational Expectations in Chinese Families." After a thorough evaluation, I have outlined several comments and suggestions below that aim to strengthen the overall quality and clarity of the manuscript:

1. While using social cognition theory is appropriate, the article should clarify how this theory specifically applies to the context of Chinese families and educational expectations.

2. The literature review is somewhat limited. It would benefit from a broader exploration of existing studies on the digital divide and educational expectations across different cultures, not just in China.

3. While the methodology is described, more detail is needed on how variables were operationalized, particularly "social interaction" and "Internet use intensity."

4. The authors should discuss potential biases in the CFPS data, such as response bias or sample representativeness, which could affect the findings.

5. The selection of control variables should be justified more thoroughly. Why were certain variables chosen, and how might omitted variables impact the results?

6. The statistical methods used (e.g., Logit model, IV-Probit) are appropriate, but the article should explain why these methods were chosen over others. Additionally, include a discussion on the assumptions underlying these models.

7. Some results, particularly in the regression tables, lack sufficient interpretation. The implications of significant coefficients should be discussed more explicitly in the text.

8. The discussion could be more critical. Address potential counterarguments or alternative explanations for the findings, especially regarding the influence of cultural factors on educational expectations.

9. While the recommendations are relevant, they could be more specific. For example, what concrete steps should policymakers take to improve Internet access and educational equity?

10. The limitations section mentions future research but could suggest specific areas for further investigation, such as longitudinal studies or comparative analyses with other countries.

11. The article would benefit from thorough proofreading to ensure consistency in formatting, especially in tables and figures. Clarity in presentation is crucial for reader comprehension.

12. Ensure that all tables and figures are referenced in the text and that their relevance is explained. Consider adding visual aids that summarize key findings for better engagement.

7. PLOS authors have the option to publish the peer review history of their article (what does this mean? ). If published, this will include your full peer review and any attached files.

**Do you want your identity to be public for this peer review?** For information about this choice, including consent withdrawal, please see our Privacy Policy .

Reviewer #3: No

---

## [Author Response · Author response to Decision Letter 3]

15 Feb 2025

Response to Reviewers' Comments

Manuscript ID: PONE-D-24-23805R2

How Internet Use Has Transformed Educational Expectations in Chinese Families

Thank you for the opportunity to revise and resubmit the manuscript (PONE-D-24-23805R2) for PLOS ONE. We sincerely appreciate the reviewers' comments. We consider all the comments an opportunity to improve the manuscript. We have now addressed all the suggested comments. The detailed response is tabulated in the following section. We have marked all the revised parts in red for your convenience.

Reviewer

I would like to express my gratitude for the opportunity to review the manuscript titled "How Internet Use Has Transformed Educational Expectations in Chinese Families." After a thorough evaluation, I have outlined several comments and suggestions below that aim to strengthen the overall quality and clarity of the manuscript:

Response:

We would like to thank you for your very positive feedback and are grateful to you for your constructive and insightful comments. We have addressed all your comments and now outline our point-by-point responses below.

Reviewer - Comment 1

While using social cognition theory is appropriate, the article should clarify how this theory specifically applies to the context of Chinese families and educational expectations.

Response 1:

Thank you for this specific comment. We have revised the relevant section to provide a clearer explanation of how this theory specifically applies to the context of Chinese families and educational expectations. Please see below for what we have revised:

In section: 2.2 Impact of Internet use on educational expectations (on pages 6-8)

…‘On the one hand, it is to observe and imitate the "educational role models" in the network.

…

The collective emphasis of Chinese families on education is closely related to the "alternative reinforcement" in social cognitive theory. When parents observe that "role models" on the internet have gained social recognition for valuing education (such as their children being admitted to prestigious schools or their family status being elevated), they will see these results as potential rewards for their own behavior and be more actively involved in their children's education.

On the other hand, there is collectivism and the rationality of educational tools.

…

The cultural sensitivity of social cognitive theory further supports its explanatory power for Chinese family education. The collectivist tradition in Chinese society has strengthened parents' motivation to achieve "family honor" through education: firstly, the normalization of social comparison. The conspicuous behavior towards their children on social networks, such as academic performance, continues to put Chinese parents under social pressure. This comparison will trigger a vicarious experience - when parents discover the achievements of other children, they will see them as potential indicators of their own children's abilities, thereby driving up educational expectations. The second is the cognitive tool of education. The Internet in China is full of cases and views that high education brings high income. This further reinforces parents' belief that education is a tool for class transition. When parents have clear expectations for the results of their investment in education (such as future economic returns for their children), they are more willing to strengthen their educational expectations. Based on this analysis, we propose the following hypothesis:

…

Reviewer - Comment 2

The literature review is somewhat limited. It would benefit from a broader exploration of existing studies on the digital divide and educational expectations across different cultures, not just in China.

Response 2:

Thank you for your insightful feedback regarding the literature review. I agree that broadening the scope to include a more diverse range of studies would strengthen the discussion. Please see below for what we have revised:

In section: 2.1 The digital divide and educational expectations (on page 6)

The impact of the digital divide on educational expectations may also be influenced by cultural backgrounds. The most typical one is gender culture. The socio-cultural norms that determine women's behavior and interests largely influence their ability to utilize information technology (21). In cultures where gender inequality is more prominent (such as Africa, India, etc.), women may have reduced access to digital technology due to social norms (22-24). Lack of self-worth, self-confidence, appropriate education, heavy household responsibilities, and other factors contribute to women's digital use disorders (25). When women take on the primary responsibility for educating their children at home, the impact of the digital divide on educational inequality will be further compounded.

Reviewer - Comment 3

While the methodology is described, more detail is needed on how variables were operationalized, particularly "social interaction" and "Internet use intensity."

Response 3:

Thank you for your feedback on the methodology section. I appreciate your suggestion to provide more detail on how key variables were operationalized. In the revised version, I have added the following clarifications:

In section: 3.2.2 Independent variable (on page 11)

Internet use: In the CFPS questionnaire, a question was designed for respondents asking 'How much leisure time do you spend online per week? (hours/week)'. We take the respondent's answer to this question as a proxy variable of Internet use intensity. The average Internet use intensity is 8.585, which means that on average, parents in the sample spend more than one hour online every day. In addition, the standard deviation of this indicator is significantly greater than the average, indicating that there is a large difference in the length of Internet use between parents.

In section: 3.2.3 Mediator variable (on pages 11-12)

In the CFPS questionnaire, a question was designed for the respondents asking “How much money did your household pay for gift giving in the last 12 months?”. We measure the level of social interaction by logarithmizing the amount of personal favors and gifts paid in the questionnaire. In the sample, the households surveyed had the highest expenditure of 80000 yuan, while the households with the lowest expenditure did not.

Reviewer - Comment 4

The authors should discuss potential biases in the CFPS data, such as response bias or sample representativeness, which could affect the findings.

Response 4:

Thank you for raising this critical point about potential biases in the CFPS data. I appreciate your attention to methodological rigor, and I have revised the manuscript to address these concerns more explicitly.

In section: 5.1 Summary

…‘Second, the limited number of years does not allow for estimating the effect of the long-term impact of internet use on educational aspirations.’…

Reviewer - Comment 5

The selection of control variables should be justified more thoroughly. Why were certain variables chosen, and how might omitted variables impact the results?

Response 5:

Thank you for your thoughtful feedback regarding the selection of control variables. I agree that a more thorough justification is necessary to ensure the robustness of the analysis. In the revised manuscript, We have added the following explanations:

In section: 3.2.4 Control variable (on pages 12-13)

…‘The control variables were chosen for the following reasons:

Child’s gender. In Asian cultures, parents may be more willing to invest more resources in education for boys, who are considered to be the main breadwinners of the family. Girls may have relatively less to invest in their education, especially if resources are limited.

Child’s age. The older a child is, the more stable his or her academic performance will be, and the more stable his or her parents' educational expectations will be.

Parental gender. With the spread of the concept of gender equality, the division of labour between parents tends to be more flexible. However, on the whole, mothers are more involved in their children's education and have higher expectations of their children's education.

Parental age. There is a correlation between the age of parents and their educational aspirations for their children, with younger parents usually favouring more open and diverse educational philosophies, while older parents may be more focused on traditional and stable educational paths.

Household register. Household registration is highly correlated with educational expectations. Parents with urban household registration have higher educational expectations for their children than those with agricultural registration (37).

Income. Economic advantages give parents more confidence and ability to support higher levels of education for their children, thus raising expectations for their children's education (38). We measure the economic advantage of households by controlling for the level of household income.

Residence. Differences in the educational aspirations of rural and urban families are both a reflection of social stratification and a result of the unequal distribution of educational resources. In China, rural parents generally wish to achieve class mobility through education.’

Reviewer - Comment 6

The statistical methods used (e.g., Logit model, IV-Probit) are appropriate, but the article should explain why these methods were chosen over others. Additionally, include a discussion on the assumptions underlying these models.

Response 6:

Thank you for your feedback regarding the statistical methods used in the article. I agree that providing a clearer rationale for the choice of methods is important for transparency and rigor. In the revised manuscript, I have added the following explanations:

In section: 4.2.3 Addressing endogenous issues (on page: 18)

‘Due to the inability of the Logit model to directly apply the instrumental variable method, following the practices of most studies, the IV-Probit model is adopted for instrumental variable estimation.’

Reviewer - Comment 7

Some results, particularly in the regression tables, lack sufficient interpretation. The implications of significant coefficients should be discussed more explicitly in the text.

Response 7:

In section: 4.1 Baseline results (on page: 14)

In columns (1) - (3), the regression coefficients of Internet use are significantly positive at the level of 1%, which indicates that the longer parents use the Internet, the greater the probability that they expect their children to go to four-year colleges. From then on, it is assumed that H1 has been validated. As the coefficients of the Logit model lack practical significance and solely indicate significance and direction, odds ratios and marginal effects were computed. Columns (4) sequentially present the odds ratios and average effects based on the estimation results in Column (3). For each additional hour of weekly Internet use, parents’ expectations of their children attending undergraduate college increase by 1.11%.

In section: 4.3 Educational expectations based on different Internet use purposes (on page: 20)

…‘Our findings reveal that when parents use the Internet for learning and work, their educational expectations for their children are the highest (Coef=0.1211, p<0.01; Coef=0.1170, p<0.01), followed by business activities and social intercourse (Coef=0.1015, p<0.01; Coef=0.0825, p<0.01). As expected, frequent online entertainment by parents does not significantly enhance their expectations for their children's education (Coef=-0.0002, p>0.1).’…

Reviewer - Comment 8

The discussion could be more critical. Address potential counterarguments or alternative explanations for the findings, especially regarding the influence of cultural factors on educational expectations.

Response 8:

Thank you for your valuable feedback on the discussion section. I agree that a more critical analysis, including potential counterarguments and alternative explanations, would enhance the depth and rigor of the article. Please see below for what we have revised:

In section: 5.1 Summary (on page: 26)

Our findings reveal new ways of linking Internet use to children's educational outcomes through parental expectations, and find that Chinese parents will be more likely to expect their children to go to college under the influence of the Internet. Of course, this influence is not always valid across cultural contexts. On the one hand, parents from Asian backgrounds will have higher academic standards and higher expectations for their children's education (48). The cognitive and learning mechanisms brought about by the Internet may be weaker for parents from cultures with low expectations for their children's education (e.g., completion of an apprenticeship). As they are not concerned with their children's education, parents may be keen to share their children's life entertainment rather than educational achievements, which may also lead to the failure of the imitation and role modelling effects that we have described in our hypothesis. On the other hand, the notion of collectivism is also an important driver. Children's achievements are often seen as the success of the family, and children's efforts and progress can bring positive social evaluation and status enhancement to the family. For cultures that emphasise individualism, parents may be more concerned with their own needs, which in turn weakens the expectation of family status enhancement through their children.

Reviewer - Comment 9

While the recommendations are relevant, they could be more specific. For example, what concrete steps should policymakers take to improve Internet access and educational equity?

Response 9:

Thank you for your constructive feedback on the policy recommendations section. I agree that providing more concrete and actionable steps would enhance the practical relevance of the article. In the revised manuscript, I have expanded the recommendations to include the following specific measures:

In section: 5.2 Findings, policy recommendations, and research limitations (on page: 29)

…‘To this end, we give the following policy recommendations: first, expand rural broadband network coverage and reduce the cost of access for households in remote areas, in response to the more significant impact of the Internet in rural households. Second, subsidise the purchase of smart terminals (e.g. tablet PCs) for low-income people e to ensure hardware accessibility. Finally, create officially certified online education communities to facilitate the sharing of experiences between urban and rural parents.’…

Reviewer - Comment 10

The limitations section mentions future research but could suggest specific areas for further investigation, such as longitudinal studies or comparative analyses with other countries.

Response 10:

Thank you for your insightful feedback on the limitations section. I agree that suggesting specific areas for future research would strengthen the article's contribution to the field. In the revised manuscript, I have expanded the limitations section to include the following specific recommendations for future investigation:

In section: 5.2 Findings, policy recommendations, and research limitations (on page: 29)

…‘Second, the limited number of years does not allow for estimating the effect of the long-term impact of internet use on educational aspirations. Third, although China’s unique educational climate and regionally differentiated Internet penetration make it an ideal case for analyzing the Internet’s impact on educational expectations, our empirical results are specific to China. Cross-country comparative analyses can subsequently be used to test the cultural universality and institutional boundary conditions of the Internet's influence on educational expectations.’…

Reviewer - Comment 11

The article would benefit from thorough proofreading to ensure consistency in formatting, especially in tables and figures. Clarity in presentation is crucial for reader comprehension.

Response 11:

Thank you for your feedback regarding the need for thorough proofreading and formatting consistency in the article. I completely agree that clarity in presentation is ess

---

## [Decision Letter · Decision Letter 3]

3 Mar 2025

How Internet Use Has Transformed Educational Expectations in Chinese Families

PONE-D-24-23805R3

Dear Dr. Lu,

We’re pleased to inform you that your manuscript has been judged scientifically suitable for publication and will be formally accepted for publication once it meets all outstanding technical requirements.

Kind regards,

Ehsan Namaziandost

Academic Editor

PLOS ONE

Additional Editor Comments (optional):

Reviewers' comments:

Reviewer's Responses to Questions

**Comments to the Author**

1. If the authors have adequately addressed your comments raised in a previous round of review and you feel that this manuscript is now acceptable for publication, you may indicate that here to bypass the “Comments to the Author” section, enter your conflict of interest statement in the “Confidential to Editor” section, and submit your "Accept" recommendation.

Reviewer #3: (No Response)

2. Is the manuscript technically sound, and do the data support the conclusions?

Reviewer #3: (No Response)

3. Has the statistical analysis been performed appropriately and rigorously? 

Reviewer #3: (No Response)

4. Have the authors made all data underlying the findings in their manuscript fully available?

Reviewer #3: (No Response)

5. Is the manuscript presented in an intelligible fashion and written in standard English?

Reviewer #3: (No Response)

6. Review Comments to the Author

Reviewer #3: The authors have diligently addressed all feedback and comments provided during the review process within the manuscript's main text. As a result, the revised version of the manuscript meets the required standards and is now deemed suitable for publication.

Good luck.

7. PLOS authors have the option to publish the peer review history of their article (what does this mean? ). If published, this will include your full peer review and any attached files.

**Do you want your identity to be public for this peer review?** For information about this choice, including consent withdrawal, please see our Privacy Policy .

Reviewer #3: **Yes: ** Reza Nemati-Vakilabad

---

## [Editor Report · Acceptance letter]

PONE-D-24-23805R3

PLOS ONE

Dear Dr. Lu,

I'm pleased to inform you that your manuscript has been deemed suitable for publication in PLOS ONE. Congratulations! Your manuscript is now being handed over to our production team.

Kind regards,

on behalf of

Dr. Ehsan Namaziandost

Academic Editor

PLOS ONE